# Multidrug Resistance in Enterococci Isolated from Cheese and Capable of Producing Benzalkonium Chloride-Resistant Biofilms

**DOI:** 10.3390/biology12101353

**Published:** 2023-10-22

**Authors:** Acácio Salamandane, Gomes Cahango, Belo Afonso Muetanene, Manuel Malfeito-Ferreira, Luísa Brito

**Affiliations:** 1LEAF—Linking Landscape, Environment, Agriculture and Food Research Center, Associate Laboratory TERRA, Instituto Superior de Agronomia, Universidade de Lisboa, Tapada da Ajuda, 1349-017 Lisbon, Portugal; gomes.silva.cahango@gmail.com (G.C.); mmalfeito@isa.ulisboa.pt (M.M.-F.); lbrito@isa.ulisboa.pt (L.B.); 2Faculdade de Ciências de Saúde, Universidade Lúrio, Campus Universitário de Marrere, Nampula 4250, Mozambique; 3CNIC—Centro Nacional de Investigação Científica, Avenida Ho Chi Min, Luanda 201, Angola; 4Faculdade de Ciências Agrárias, Universidade Lúrio, Campus Universitário de Unango, Sanga 3300, Mozambique; bmuetanene@unilurio.ac.mz

**Keywords:** *Enterococci*, vancomycin resistance, multidrug resistance, virulence, benzalkonium chloride (BAC), minimum inhibitory concentration (MIC), biofilm eradication

## Abstract

**Simple Summary:**

*Enterococci* participate in cheese production, either spontaneously or when added to milk as starter cultures. These bacteria contribute to the development of the flavor, aroma, and texture of the cheese, as well as to its preservation. Despite their potential, some strains of *E. faecium* and *E. faecalis*, normal inhabitants of the intestinal tract, can cause infections. Therefore, their presence in food and water has been used as an indicator of fecal contamination. The results presented here show high levels of resistance to the antibiotic vancomycin and multidrug resistance. Some of these strains had virulence genes. These same enterococci were also capable of producing biofilms resistant to the disinfectant benzalkonium chloride. These findings highlight the potential risk of the presence of *E. faecium* and *E. faecalis* in cheese and the importance of implementing efficient control measures to guarantee the safety of dairy products.

**Abstract:**

This study aimed to investigate *enterococci* recovered from eight Portuguese cheeses made with raw ewe’s milk, regarding antibiotic resistance, virulence genes, minimum inhibitory concentration (MIC) of benzalkonium chloride (BAC), biofilm formation capacity, and biofilm eradication (MBEC) by BAC. Antimicrobial resistance against seven antibiotics of five groups was evaluated using the disk diffusion method. The presence of the genes that encode resistance to the antibiotics penicillin (*blaZ*), erythromycin (*ermA*, *ermB,* and *ermC*), vancomycin (*vanA* and *vanB*), aminoglycoside (*aac*(*6*′)*-Ie-aph*(*2*″)*-Ia*), and β-lactam (*pbp5*) and the genes that encode virulence factors, *frsB*, *cylA*, *gelE*, *esp,* and *agg*, were investigated via multiplex PCR. The susceptibility of planktonic cells to BAC was evaluated by the MIC and MBC values of the isolates, using the broth microdilution method. To assess the biofilm-forming ability and resistance of biofilms to BAC, biofilms were produced on stainless steel coupons, followed by exposure to BAC. The results showed a high resistance to the antibiotics vancomycin (87.5%), erythromycin (75%), tetracycline (50%), and penicillin (37.5%). Multidrug resistance was observed in 68.8% of the isolates. Genes encoding the virulence factors FrsB (*frsB*) and gelatinase E (*gelE*) were detected in all isolates. The *esp* and *cylA* genes were found in 56.3% and 37.5% of the isolates, respectively. All isolates exhibited a biofilm-forming ability, regardless of incubation time and temperature tested. However, after 72 h at 37 °C, *E. faecium* and *E. faecalis* biofilms showed significant differences (*p* ≤ 0.05). Although most isolates (62.5%) were susceptible to BAC (MIC ≤ 10 mg/L), biofilms of the same isolates were, generally, resistant to the higher concentration of BAC (80 mg/mL) tested. This study using *Enterococcus* isolates from a ready-to-eat food, such as cheese, reveals the high percentages of vancomycin resistance and multidrug resistance, associated with the presence of virulence genes, in isolates also capable of producing biofilms resistant to BAC, an important active ingredient of many disinfectants. These results emphasize the need for effective control measures to ensure the safety and quality of dairy products.

## 1. Introduction

*Enterococcus faecium* and *Enterococcus faecalis* are found in the gastrointestinal tract of humans and animals [1]. These microorganisms have traditionally been used, directly or indirectly, in cheese production as starter cultures, due to their significant contribution to the development of flavor, aroma, and texture, as well as their role in protecting cheese against spoilage [2]. In addition to their role in cheese production, *E. faecium* and *E. faecalis* have also been explored for their potential as probiotics [2,3]. The use of probiotics in foods and dietary supplements has gained increasing attention in recent years due to their potential to promote gut health, modulate the immune system, and prevent or treat various diseases [4,5,6].

Several studies have investigated the probiotic properties of *E. faecium* and *E. faecalis* and their safety for human consumption [1,3,4,7]. *E. faecium* exhibits probiotic activity, improving the composition of the intestinal microbiota, enhancing the immune response, and reducing the risk of gastrointestinal infections [8,9]. It has also been reported that *E. faecium* has a beneficial effect on the cardiovascular system, reducing cholesterol levels and improving blood pressure [10,11,12]. Likewise, *E. faecalis* has also been explored for its potential as a probiotic. Some strains of *E. faecalis* have been found to exhibit antimicrobial activity against pathogenic bacteria and increase the production of cytokines and immunoglobulins [13]. Furthermore, *E. faecalis* has also been reported to have a beneficial effect on the gastrointestinal tract by reducing inflammation and promoting the growth of beneficial bacteria [14,15].

Despite their potential as starter and probiotics strains, it is important to note that some strains of *E. faecium* and *E. faecalis*, which are normal inhabitants of the intestinal tract, can cause various infections under certain conditions, including cystitis, pyelonephritis, endocarditis, and abdominal, urinary tract and pelvic infections [16,17,18,19]. Furthermore, its presence in food products and water is an indicator of fecal contamination [20,21,22]. 

*Enterococcus* has been associated with antibiotic resistance and the production of virulence factors, which may pose a health risk [23,24]. Antibiotic resistance is a growing threat to public health, with the use of antibiotics in animal husbandry identified as a significant contributor to the emergence and spread of antibiotic-resistant bacteria. *Enterococcus* species have been shown to be highly resistant to a wide range of antibiotics, including vancomycin, which is considered a last-resort antibiotic for the treatment of severe infections [4,25,26].

Several studies have shown various strains of *E. faecium* and *E. faecalis* with the ability to form biofilms on various surfaces, including stainless steel, polyvinyl chloride, and polystyrene [27,28]. The formation of biofilms with *E. faecium* and *E. faecalis* in cheese production can have both positive and negative effects. On the one hand, biofilms can contribute to the development of desirable sensory properties of the cheese, such as flavor and texture [29]. On the other hand, biofilms can also lead to the persistence and spread of pathogenic bacteria, which can compromise cheese safety [28,30,31].

This study aimed to evaluate the potential risk of the presence of *E. faecium* and *E. faecalis* in Portuguese cheese made with raw ewe’s milk, related to the presence of virulence genes, resistance to antibiotics, and resistance of the cells to benzalkonium chloride (BAC) disinfectant in the planktonic and in the biofilm state.

## 2. Materials and Methods

### 2.1. Enterococcus Isolates from Cheeses

This study used 16 *Enterococcus* spp. isolates (8 *E. faecalis* and 8 *E. faecium*), recovered from eight semi-soft ripened Portuguese cheeses produced in the north of Portugal, through the coagulation of raw ewe’s milk. The cheeses were collected in November 2022, in a retail establishment in Lisbon, or ordered online from the producer and delivered in Lisbon. The cheeses were received at the laboratory, refrigerated (4–8 °C), and analyzed within less than 24 h. Aliquots of 10 g of each cheese were removed by cutting radially and vertically into three nearly equidistant slices (which included approximately equal amounts of material from the inner and outer parts of the cheese) and placed in sterile Bag Filters (Interscience, Saint Nom la Brétèche, France). The Bag Filters were filled with 90 mL of sterile Ringer’s solution (SR) (Biokar Diagnostics, Beauvais, France) and homogenized in a paddle blender (400 Circulator, International PBI, Milan, Italy) at 85 rpm for 2 min. From each suspension, successive decimal dilutions were performed in RS according to the ISO Standard (ISO 6887, 2017) [32,33]. Immediately afterwards, 100 μL of each selected dilution was inoculated in duplicate onto plates of Compass Enterococcus Selective Agar (Biokar Diagnostics, Beauvais, France) and incubated at 44 °C for 24 h. 

For each cheese, two well-isolated and characteristic colonies were chosen from the chromogenic plates for biochemical (Gram staining, catalase, and oxidase tests) and molecular tests. For molecular identification, multiplex PCR with the primers Ent1 (TACTGACAAACCATTCATGATG) and Ent2 (AACTTCGTCACCAACGCGAAC) (112 bp) for *Enterococcus* spp., ddlE1 (ATCAAGTACAGTTAGTCTT) and ddlE2 (ACGATTCAAAGCTAACTG) (941 bp) for *E. faecalis*, and ddlF1 (GCAAGGCTTCTTAGAGA) and ddlF2 (CATCGTGTAAGCTAACTTC) (550 bp) for *E. faecium* were performed as described by Rocha et al. (2022) [34].

Confirmation of the identification of the isolates was performed, targeting the 16S rRNA gene (800 bp) with the universal primers Bac27F (5′-AGAGTTTGGATCMTGGCTCAG-3′) and Univ1492R (5′-CGGTTACCTTGTTACGACTT-3′) [35]. 

The amplified products were sequenced (STAB VIDA, Caparica, Portugal) and the resulting sequences were submitted to Blastn using the megablast algorithm against Reference RNA sequences database (RefSeq RNA) from the National Center of Biotechnology Information (NCBI) for isolate identification. Afterwards, the sequences were used to construct the phylogenetic tree (Figure 1).

### 2.2. Antibiotic Resistance Profile, Virulence, and Resistance Genes

The antibiotic susceptibility profile to five antibiotic groups was evaluated using the disk diffusion method, according to the Clinical and Laboratory Standards Institute guidelines [36]. The following seven antibiotics were used: gentamicin (GEN, 120 μg), penicillin (P, 10 U), erythromycin (ERY, 15 μg), tetracycline (TET, 30 μg), fosfomycin (FOS, 200 μg), rifampicin (RD, 5 μg), and vancomycin (VAN, 30 μg).

The presence of the virulence genes *fsrB*, *cylA*, *gelE*, *esp,* and *agg* was investigated. The respective primers, their concentrations, and the respective multiplex PCR conditions were those previously used by Semedo-Lemsaddek et al., 2021 [37]. 

The presence of genes encoding resistance to penicillin (*blaZ*), erythromycin (*ermA*, *ermB* and *ermC*), vancomycin (*vanA* and *vanB*), aminoglycoside (*aac*(*6*′)*-Ie-aph*(*2*″)*-Ia*), and β-lactam (*pbp5*) was screened via multiplex PCR. The respective primers used, concentrations, product sizes, and PCR conditions were previously described by Salamandane et al., 2022 [38] (*blaZ*, *ermA*, *ermB* and *ermC*; *vanA,* and *vanB*) and by Semedo-Lemsaddek et al., 2021 [37] (*aac*(*6*′)*-Ie-aph*(*2*″)*-Ia* and *pbp5*).

### 2.3. Evaluation of the Susceptibility to Benzalkonium Chloride (BAC)

The minimum inhibitory concentration (MIC) and the minimum bactericidal concentration (MBC) of BAC (Sigma–Aldrich, St. Louis, MO, USA) against *E. faecalis* and *E. faecium* were determined in Mueller–Hinton broth (Biolife, Milano, Italy) at 37 °C and in triplicate, at least in two independent assays. We used 96-well polystyrene microtiter plates (Greiner Bio-One, Frickenhausen, Germany) to utilize the microdilution broth method using two-fold dilutions of BAC [36].

For MIC determination, BAC was used in the range of 1.25–80 mg/L. Microplates were inoculated with 10^4^ CFU per well, in a total volume of 200 µL, and incubated aerobically for 24 h. After the incubation period, the wells that showed visible turbidity in contrast to the non-inoculated controls were considered positive results. When at least two of the three replicates showed turbidity, the result was considered positive (growth). When only one or none of the replicates showed turbidity, the result was considered negative (absence of growth). The MIC was defined as the lowest concentration of BAC that inhibited the visible growth of the isolates in contrast to the non-inoculated controls. Isolates with MIC ≤ 10 mg/L were considered susceptible to BAC [39].

The MBC of BAC was determined after the MIC determination, namely, from all the wells that did not show visible bacterial growth after 24 h incubation at 37 °C, 100 µL was removed, inoculated onto TSA-YE, Trypto–Casein–Soy agar plates (Biokar Diagnostics, Beauvais, France), and incubated at 37 °C for 24 h. MBC was defined as the lowest concentration of BAC capable of killing 99.9% of the bacterial population (3 Log reduction) [40,41].

### 2.4. Evaluation of Biofilm-Forming Ability

The evaluation of the biofilm-forming ability of the isolates was performed based on the removal of cells from the biofilms formed on the surface of stainless-steel coupons (SSCs), 1 cm × 1 cm and 1 mm thick, type 304, 2B finish (Metalurgica Quinacorte, Lda, Lousa, Portugal), as previously described [40,41]. Briefly, each coupon was immersed in a well of a 24-well polystyrene microplate containing 1.5 mL of inoculum in TSB (about 10^8^ cfu/mL). Microplates were sealed with Parafilm^®^ and incubated at 37 °C for 24, 48, or 72 h, or at 15 °C for 72 h. After incubation, the coupon was rinsed with RS (1 mL for each side) to remove planktonic cells and replaced in a new 24-well microplate containing 20 glass beads (Ø = 3 mm) per well. Another 30 glass beads and 1 mL of RS were placed on top of each coupon.

The 24-well microplate was shaken on a microplate shaker (Tittertek DSG, Flowlabs, Berlin, Germany) for 5 min at maximum speed to disaggregate the biofilm cells. Subsequently, decimal dilutions were performed and TSA-YE plates were inoculated through spreading. The CFU count assessment was performed after 24 h and confirmed after 48 h of incubation at 37 °C. After calculating the total adhesion surface area, the biofilm formation capacity of the isolates was expressed in log CFU/cm^2^. At least two biological replicates were performed with two technical replicates each.

### 2.5. Minimum Biofilm Eradication Concentration (MBEC) of BAC

To evaluate the ability of BAC to remove *Enterococci* biofilms, biofilms were produced as described above and, after incubation, coupons were rinsed with RS (1 mL for each side) to remove planktonic cells and placed in a new 24-well microplate with different concentrations of BAC (10, 20, 40, and 80 mg/L). Coupons were exposed to BAC for 5 min at room temperature. After this period, the coupons were removed, rinsed with RS, as previously described, and placed in a new 24-well microplate containing 1 mL of Dey–Engley neutralizing broth (D/E) solution (Difco Laboratories, New Jersey, NJ, USA) per well and incubated at room temperature for 5 min. After this period of contact with D/E for neutralization, coupons were replaced in a new 24-well microplate with 20 glass beads per well. Another 30 glass beads and 1 mL RS were placed on top of each coupon. 

The 24-well microplate was shaken on a microplate shaker (Tittertek DSG, Flowlabs, Berlin, Germany) for 5 min at maximum speed to disaggregate adhered cells. Subsequently, from each well (coupon exposed to BAC or control coupon not exposed), decimal dilutions were performed and plates of TSA-YE inoculated through spreading. The CFU count was performed after 24 h and confirmed after 48 h of incubation at 37 °C. The treatment was considered effective if a 3 Log reduction (difference between log CFU/cm^2^ of non-exposed (control) and exposed SSC to BAC) was observed. At least two biological replicates were performed with two technical replicates each.

### 2.6. Data Analysis and Interpretation 

For the evaluation of the antimicrobial resistance profile of the isolates, the inhibition halos were measured (millimeter) and compared with those described in the CLSI (2021) [36]. Isolates were considered non-susceptible to a given antibiotic when they showed intermediate or full resistance according to the CLSI clinical breakpoints. Multidrug resistance was considered as non-susceptibility to at least one agent in three or more antimicrobial categories [42,43].

Before performing the ANOVA, values obtained for the viable cell count in biofilms (CFU/cm^2^) were transformed into log CFU/cm^2^. Compliance with data normality was determined using the Shapiro–Wilk test. The homogeneity of variance was determined through the Bartlett test. ANOVA was then performed, and the Scott–Knott test was used to compare mean of different time–temperature binomials evaluated in this study. All statistical analyses were performed at 5% in the R programming language (R, 2020).

## 3. Results

### 3.1. Antibiotic Resistance Profile, Antibiotic Resistance Genes, and Virulence Genes

The results of the characterization of the isolates regarding the antibiotic resistance profile and presence of antibiotic resistance and virulence genes are presented in Table 1. None of the isolates showed resistance to rifampicin or fosfomycin. Resistance to vancomycin, tetracycline, and erythromycin were more frequent in *E. faecalis*. In both, the most frequent resistances were to vancomycin, erythromycin, and penicillin (Table 1). Vancomycin was almost complete (7/8), respectively, for *E. faecalis* and *E. faecium*. Multidrug resistance was observed both in *E. faecium* (5/8) and *E. faecalis* (6/8) isolates (Table 1).

Vancomycin resistance genes were found in all *E. faecalis* isolates. Additionally, the occurrence of erythromycin resistance genes, specifically *ermC*, was observed in seven out of the eight *E. faecalis* isolates (Table 1). Among the *E. faecium* isolates, vancomycin resistance genes were detected in seven out of the eight isolates, while erythromycin resistance genes were identified in four isolates (Table 1). The gene conferring resistance to β-lactam antibiotics, *pbp5*, was found in five *E. faecium* isolates and in one *E. faecalis* isolate. All isolates of *E. faecalis* and six out of the eight *E. faecium* isolates carried more than one type of antibiotic resistance gene.

The virulence genes *fsrB* and *gelE* were found in all the isolates (Table 1). The *esp* gene was detected in four isolates of *E. faecalis* and in five isolates of *E. faecium*. Three isolates of *E. faecalis* showed the *agg* gene (Table 1) and the gene *cylA* was present in four isolates of both *E. faecalis* and *E. faecium*.

### 3.2. Minimum Inhibitory Concentration (MIC) and Minimum Bactericidal Concentration (MBC) of Benzalkonium Chloride (BAC)

For both *E. faecium* and *E. faecalis* species, the MIC values of BAC ranged from 5 to 20 mg/L (Table 2). Fifty percent of the *E. faecalis* isolates presented MIC equal to 5 mg/L, and 25% showed MIC equal to 10 and 20 mg/L, respectively. In *E. faecium*, 50% and 37.5% of the isolates showed MIC values of 20 and 10 mg/L, respectively (Table 2). Ten in sixteen (62.5%) isolates were susceptible to BAC (MIC ≤ 10 mg/L). All *E. faecium* isolates showed MBC values twice as high as MIC values (Table 2). On the other hand, 37.5% of *E. faecalis* showed MBC values equal to MIC.

### 3.3. Biofilm-Forming Ability According to Incubation Time and Temperature

Regardless of the incubation time and temperature, all isolates showed the ability to form biofilms. In fact, biofilms were produced at 15 °C during 72 h and at 37 °C during 24 h, 48 h, and 72 h, respectively (Table 3). In *E. faecalis*, biofilms produced at 37 °C varied between 6.4 Log CFU/cm^2^ (24 h) and 9.04 Log CFU/cm^2^ (72 h) (Table 3). In *E. faecium*, biofilms produced at 37 °C varied between 6.15 and 8.18 Log CFU/cm^2^ (24 h and 72 h, respectively) (Table 3). For each species, after 72 h at 37 °C, significant differences were observed in the biofilm-forming ability of the isolates (Table 3). When the isolates were incubated at 15 °C for 72 h, a great heterogeneity in the biofilm production of the isolates was observed. In these conditions, the highest biofilm production (7.87 Log CFU/cm^2^) was observed in the *E. faecalis* isolate R66 and the lowest biofilm production (6.69, 6.82, and 6.67 CFU/cm^2^) was observed in *E. faecium* isolates (R12, R106, and R131, respectively) (Table 3).

### 3.4. Assessment of the Minimum Biofilm Eradication Concentration by BAC (MBEC)

The evaluation of biofilm eradication by BAC was performed with biofilms produced at 37 °C for 48 h. The BAC concentration ranged from 10 mg/L to 80 mg/L (twice the maximum value obtained for MBC). Six isolates (R38, R46, R17, R44, R131, and R150) showed significant differences in log CFU/cm^2^ reduction among all treatments (*p* < 0.05) (Figure 2). In four isolates (R37, R66, R12, and R29), no significant differences were observed between the last two highest concentrations (*p* > 0.05).

In *E. faecalis*, the greatest reduction (3.78 Log CFU/cm^2^) was observed in R38, with 80 mg/L of BAC, followed by isolate R108, which showed a reduction of 2.56 Log CFU/cm^2^. The lowest reduction observed with this concentration (1.24 Log CFU/cm^2^) corresponded to isolate R64. 

In the *E. faecium,* the greatest reduction (4.4 Log CFU/cm^2^) was observed in R17 with the application of 80 mg/mL of BAC. This isolate (R17) showed the greatest reductions with all BAC concentrations used, being the most susceptible biofilm to BAC (Figure 2). On the other hand, the lowest reduction observed (1.34 Log CFU/cm^2^) with the maximum BAC concentration was in another *E. faecium* isolate, R44. This was also the least susceptible biofilm to the four BAC concentrations tested. Only two isolates (R38 and R17) achieved a reduction in the effective defend threshold (3 Log reduction).

## 4. Discussion

*Enterococci* are one of the most common groups of bacteria present in cheese, mainly due to their significant contribution to the development of cheese flavor, aroma, and texture, as well as their role in protecting the cheese against spoilage [2]. Like other lactic acid bacteria, *enterococci* are resistant to the harsh environmental conditions of fermentation as well as the storage conditions of fermented foods. These characteristics make *E. faecium* and *E. faecalis* frequently used as probiotic strains in several supplements and in dietary foods and beverages [2,3]. However, the use of *enterococci* in homemade fermented foods and beverages has raised concerns due to the potential occurrence of antibiotic resistance genes and the possibility of a horizontal transfer of these genes [44].

In the present study, 87.5% of *enterococci* collected from cheese made with raw ewe’s milk were resistant to vancomycin (VRE). Furthermore, all VRE were multidrug-resistant and had the *vanA* gene. The isolates also showed a high level of resistance to erythromycin (75%) and tetracycline (50%). Similar results were found with *enterococci* isolated from sheep and goat milk cheeses. *E. faecium* showed 100% resistance to vancomycin, and *E. faecalis* showed 85.7% resistance to vancomycin and 71.4% resistance to erythromycin [45]. Vancomycin resistance is more frequent in *E. faecalis*, whereas most *E. faecium* strains are highly resistant to beta-lactams and aminoglycosides such as erythromycin [46].

In *enterococci*, *vanA* is one of the most important genes involved in the regulation and expression of vancomycin resistance. This gene and other VRE genes (*vanR*, *vanS*, *vanH*, *vanX,* and *vanZ*), are located on the *E. faecium* transposon Tn1546, which often resides on a plasmid [46]. The expression of these genes results in the synthesis of abnormal peptidoglycan precursors that end in D-Ala–D-lactate instead of D-Ala–D-Ala [46,47]. As this gene is plasmid-mediated, it is likely that the vancomycin resistance found in *E. faecium* was a consequence of a horizontal gene transfer in the sheep grazing environment or during the cheese production process. On the other hand, the cheeses used in this study, which were the source of the isolates analyzed, are from the same geographic region. Therefore, vancomycin resistance genes may be circulating in the environment.

The erythromycin resistance genes can also be transmitted under identical circumstances. In *enterococci*, erythromycin resistance is associated with the presence of erythromycin ribosome methylase (*erm*) genes [48], such as *ermA*, *ermB,* and *ermC*. These genes were initially observed in the Tn554 transposon on the *Staphylococcus aureus* chromosome [49]. Tetracycline resistance was found in 75% of *E. faecalis* and in 25% of *E. faecium* isolates. A similar result was found in *enterococci* recovered from Portuguese cheeses from Azeitão and Nisa (21 to 97%) [34]. In the present study, all tetracycline-resistant phenotypes were associated with the presence of the *tetM* gene. This gene, highly prevalent among enterococcal species, is located mainly on the bacterial chromosome and often linked to conjugative transposons associated with the Tn916/Tn1545 family [42].

The presence of the gelatinase (*gelE*) and gelatinase regulator (*fsrB*) genes in all isolates used in this study emphasizes the importance of these strains in terms of public health. Gelatinase, the product of the *gelE* gene, is an important virulence factor of *Enterococcus* spp. It contributes to tissue invasion, immune evasion, biofilm formation, and host cell modulation. All these aspects contribute to enhancing the ability of *Enterococcus* to cause infections and establish persistent colorizations [19,43]. 

Esp is considered a key virulence factor of *E. faecalis* and *E. faecium*. Its presence increases the pathogenic potential of these bacteria and contributes to the severity of associated infections [50,51]. Esp-positive strains have been associated with increased virulence in clinical settings [50]. In a similar study with raw milk cheeses in Poland, *gelE* was the most prevalent gene (88.9%), followed by *esp* (36.1%) and *fsrB* (22.2%) [52].

All *E. faecalis* and *E. faecium* isolates showed biofilm formation ability. However, the greatest biofilm production occurred at 37 °C for 72 h, with the biofilm formation ability of *E. faecalis* being significantly higher than that of *E. faecium* (*p* ≤ 0.05) in these conditions. This good ability to form biofilms corroborates other studies that establish the occurrence of genes such as *gelE*, *fsrB*, *spe,* and *agg* as factors that enhance biofilm production in *Enterococcus* species [28]. In addition to the presence of these virulence genes, it is important to highlight that all isolates were also resistant to vancomycin. Indeed, several studies have reported vancomycin-resistant *enterococci* as strong biofilm producers [28,53]. 

Benzalkonium chloride is a quaternary ammonium compound widely used as a disinfectant in the food industry. With its potent biocidal properties, BAC effectively combats a wide range of microorganisms, including bacteria, viruses, and fungi [54]. Among the *E. faecalis* isolates, six out of eight were susceptible to BAC (MIC ≤ 10 mg/L) [39]. Two isolates exhibited MIC ≥ 10 mg/L (BAC resistance) [39]. Interestingly, despite this resistance, the MBC/MIC ratio for these isolates was 1. This result suggests that achieving a bactericidal effect may not require excessively high doses of this disinfectant, even for the isolates with higher MIC values [55]. In *E. faecium,* six out of eight isolates had MIC ≥ 10 mg/L, and the MBC/MIC ratio for all *E. faecium* isolates was 1.

Although BAC showed significant biocidal activity in this study, it is worth noting that the maximum concentration tested, 80 mg/mL, was not sufficient to eradicate biofilms in most isolates. Among the *E. faecalis* isolates, one of the treatments demonstrated a decrease ≥3 Log CFU/m^2^. However, significant differences were observed among treatments in the majority of the isolates, indicating that higher concentrations of BAC led to improved treatment efficiency. In the case of the *E. faecium* isolates, only one isolate (R17) showed a reduction greater than 3 Log CFU/cm^2^. Furthermore, a significant difference (*p* ≤ 0.05) in the reduction was observed with an increasing concentration of BAC with five/eight isolates. In a similar study, Barroso et al., 2020 [56] concluded that the BAC-resistant/-sensitive phenotype of the isolates did not dictate the susceptibility of their biofilm counterparts.

*Enterococcus faecalis* and *Enterococcus faecium* have gained significance in recent decades as leading opportunistic pathogens causing nosocomial infections [57]. The occurrence in cheese-processing environments of persistent strains of *Enterococcus* that are resistant to multiple antibiotics, carrying virulence genes and capable of forming disinfectant-resistant biofilms, represents a significant threat to human health. This can lead to an increase in cases of infections, as these isolates may spread to the community and clinical settings during the processing, distribution, and/or consumption of the cheese.

## 5. Conclusions

This study characterized isolates of *E. faecium* and *E. faecalis* obtained from Portuguese cheese made with raw ewe’s milk. The results show high levels of VRE and of virulence genes, associated with biofilm-forming ability and resistance of biofilms to BAC, a widely used disinfectant agent. Particularly relevant was the fact that the BAC-resistant or -sensitive phenotype of the strains in the planktonic state did not dictate the susceptibility of their biofilm counterparts. Given that biofilms contribute to the persistence of bacteria throughout the food chain and represent an important source of food contamination, these results emphasize the importance of efficient cleaning of equipment and utensils used in cheese production. On the other hand, more effective control measures are needed, particularly regarding animal management and milking conditions, to guarantee the safety and quality of dairy products.

## Figures and Tables

**Figure 1 biology-12-01353-f001:**
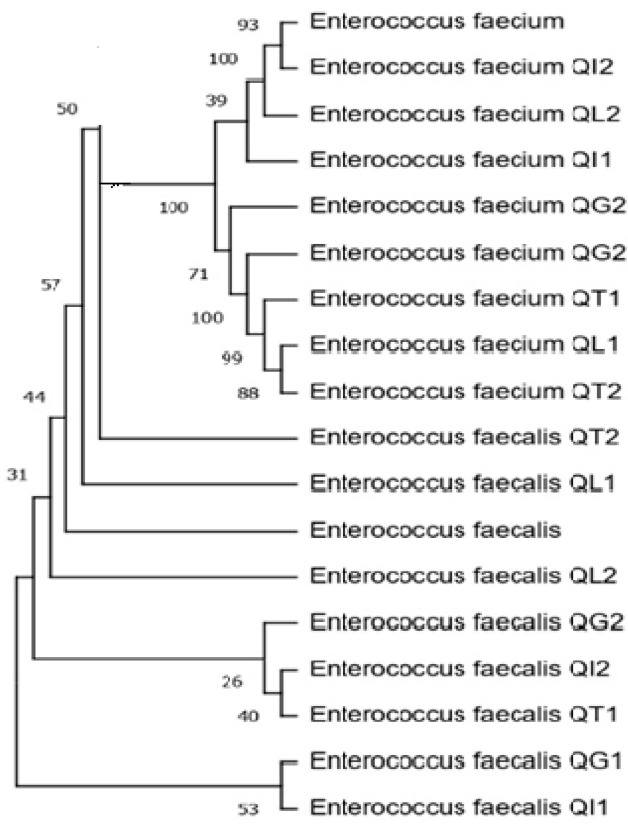
Dendrogram of the sequences of amplified fragments of the 16S rRNA genes recovered from 16 cheese isolates (the cheese code corresponding to the isolate’s origin is provided ahead of the species identification). The tree was constructed using the neighbor-joining method. Evolutionary distances were calculated using the maximum composite likelihood method.

**Figure 2 biology-12-01353-f002:**
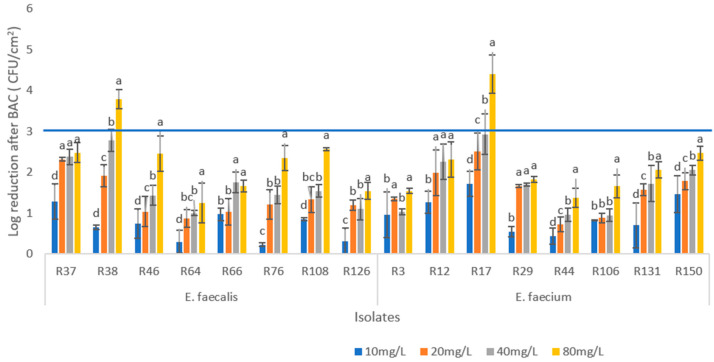
Logarithmic reduction of colony-forming units (log reduction CFU/cm^2^) after exposure of *enterococci* biofilms to four concentrations of benzalkonium chloride (BAC) (10, 20, 40, and 80 mg/mL) for 5 min. For each isolate, different letters in the columns indicate significant differences (*p* < 0.05) between average values. Horizontal blue line indicates log reduction threshold for minimum biofilm eradication concentration (MBEC).

**Table 1 biology-12-01353-t001:** Antibiotic resistance profile, antibiotic resistance genes, and virulence genes in *Enterococcus* spp. isolates from raw ewe’s milk cheese.

Isolate Code	Species	Antibiotic Resistance		Virulence Genes
		Phenotype	Genotype	
**R37**	** *E. faecalis* **	RD, ERY	*ermC, vanA*	*FsrB* *, gelE, cylA*
**R38**	** *E. faecalis* **	**TET, ERY, VAN**	** *ermC, tetM, vanA* **	** *fsrB* *, gelE, esp, agg* **
R46	** *E. faecalis* **	VAN	*vanA* *, vanB*	*FsrB* *, gelE*
**R64**	** *E. faecalis* **	**TET, RD, ERY, VAN**	** *ermC, tetM, vanA, pbp5* **	** *fsrB* *, gelE, cylA, esp* **
**R66**	** *E. faecalis* **	**P, TET, RD, ERY, VAN**	** *ermC, tetM, vanA, blaZ* **	** *FsrB* *, gelE, cylA* **
**R76**	** *E. faecalis* **	**P, TET, ERY, RD, VAN**	** *ermC, vanA, blaZ* **	** *fsrB* *, gelE, esp, agg* **
**R108**	** *E. faecalis* **	**TET, RD, ERY, VAN**	** *tetM, ermC, vanA* **	** *fsrB* *, gelE, cylA, esp, agg* **
**R126**	** *E. faecalis* **	**TET, ERY, VAN**	** *ermC, tetM, vanA* **	** *fsrB* *, gelE, cylA* **
**R3**	** *E. faecium* **	**TET, ERY, VAN**	** *ermA, ermC, tetM, vanA* **	** *fsrB* *, gelE, cylA, esp* **
R12	*E. faecium*	RD	*pbp5*	*fsrB* *, gelE, cylA*
**R17**	** *E. faecium* **	**GEN, ERY, VAN**	** *ermA, ermC, tetM, vanA* **	** *fsrB* *, gelE, esp* **
R29	*E. faecium*	P, VAN	*vanA*	*fsrB* *, gelE*
R44	*E. faecium*	ERY, VAN	*vanA* *, pbp5*	*fsrB* *, gelE, cylA, esp*
**R106**	** *E. faecium* **	**P, RD, ERY, VAN**	** *vanA* *, blaZ, pbp5* **	** *fsrB* *, gelE, esp* **
**R131**	** *E. faecium* **	**P, GEN, RD, VAN**	** *ermC, vanA, pbp5* **	** *fsrB* *, gelE* **
**R150**	** *E. faecium* **	**P, TET, ERY, VAN**	** *ermC, vanA, tetM, pbp5* **	** *fsrB* *, gelE, cylA, esp* **

Gentamicin (GEN, 120 μg), penicillin (P, 10 U), erythromycin (ERY, 15 μg), tetracycline (TET, 30 μg), fosfomycin (FOS, 200 μg), rifampicin (RD, 5 μg), and vancomycin (VAN, 30 μg). Multidrug-resistant isolates are signed in bold.

**Table 2 biology-12-01353-t002:** MIC and MBC values of BAC for *enterococci* isolates and the respective MBC/MIC.

Isolate Code	Species	MIC (mg/L)	MBC (mg/L)	MBC/MIC
R37	*E. faecalis*	10	20	2
**R38**	** *E. faecalis* **	**20**	**20**	**1**
R46	*E. faecalis*	20	20	1
**R64**	** *E. faecalis* **	**10**	**10**	**1**
**R66**	** *E. faecalis* **	**5**	**10**	**2**
**R76**	** *E. faecalis* **	**5**	**10**	**2**
**R108**	** *E. faecalis* **	**5**	**10**	**2**
**R126**	** *E. faecalis* **	**5**	**10**	**2**
**R3**	** *E. faecium* **	**5**	**10**	**2**
R12	*E. faecium*	20	40	2
**R17**	** *E. faecium* **	**20**	**40**	**2**
R29	*E. faecium*	20	40	2
R44	*E. faecium*	20	40	2
**R106**	** *E. faecium* **	**10**	**20**	**2**
**R131**	** *E. faecium* **	**5**	**10**	**2**
**R150**	** *E. faecium* **	**5**	**10**	**2**

Multidrug-resistant isolates are signed in bold.

**Table 3 biology-12-01353-t003:** Biofilm-forming ability of *enterococci* under different temperature conditions and incubation times.

Isolate Code	Species	Biofilm-Forming Ability (Log CFU/cm^2^)				*p*-Value
		15 °C, 72 h	37 °C, 24 h	37 °C, 48 h	37 °C, 72 h	
R37	*E. faecalis*	7.66 ± 0.31 ^Bb^	6.40 ± 0.09 ^Dd^	7.07 ± 0.06 ^Cb^	8.82 ± 0.09 ^Aa^	3.7 × 10^−6^
R38	*E. faecalis*	7.13 ± 0.05 ^Cd^	7.40 ± 0.04 ^Ba^	7.29 ± 0.02 ^Bb^	8.82 ± 0.10 ^Aa^	9.99 × 10^−9^
R46	*E. faecalis*	7.25 ± 0.01 ^Cc^	7.13 ± 0.14 ^Cb^	7.60 ± 0.16 ^Ba^	8.89 ± 0.07 ^Aa^	1.1 × 10^−6^
R64	*E. faecalis*	6.92 ± 0.16 _Cd_	7.10 ± 0.05 ^Cb^	7.92 ± 0.07 ^Ba^	8.89 ± 0.07 ^Aa^	1.3 × 10^−7^
R66	*E. faecalis*	7.87 ± 0.08 ^Ba^	6.76 ± 0.28 ^Cc^	7.87 ± 0.09 ^Ba^	8.78 ± 0.14 ^Aa^	1.8 × 10^−5^
R76	*E. faecalis*	7.06 ± 0.14 ^Cd^	6.50 ± 0.01 ^Dc^	7.75 ± 0.10 ^Ba^	8.91 ± 0.06 ^Aa^	2.57 × 10^−8^
R108	*E. faecalis*	7.52 ± 0.05 ^Cb^	7.28 ± 0.21 ^Ca^	7.81 ± 0.04 ^Ba^	8.89 ± 0.08 ^Aa^	3.1 × 10^−6^
R126	*E. faecalis*	6.97 ± 0.07 ^Cd^	6.72 ± 0.11 ^Cc^	7.70 ± 0.16 ^Ba^	9.04 ± 0.18 ^Aa^	7.8 × 10^−7^
R3	*E. faecium*	7.28 ± 0.13 ^Bc^	6.21 ± 0.08 ^Cd^	7.90 ± 0.06 ^Aa^	8.15 ± 0.46 ^Ab^	0.00019
R12	*E. faecium*	6.69 ± 0.09 ^Be^	6.65 ± 0.11 ^Bc^	7.41 ± 0.38 ^Ab^	8.01 ± 0.77 ^Ab^	0.0418
R17	*E. faecium*	7.31 ± 0.15 ^Bc^	7.48 ± 0.34 ^ABa^	7.49 ± 0.45 ^ABb^	7.96 ± 0.61 ^Ab^	0.0486
R29	*E. faecium*	7.51 ± 0.02 ^Ab^	6.38 ± 0.11 ^Bd^	7.18 ± 0.01 ^Bb^	7.70 ± 0.27 ^Ab^	9.3 × 10^−5^
R44	*E. faecium*	7.21 ± 0.11 ^Bc^	6.93 ± 0.05 ^Bb^	7.84 ± 0.21 ^Aa^	8.18 ± 0.45 ^Ab^	0.00415
R106	*E. faecium*	6.82 ± 0.04 ^Be^	6.98 ± 0.04 ^Bb^	7.78 ± 0.02 ^Aa^	7.90 ± 0.05 ^Ab^	0.0418
R131	*E. faecium*	6.67 ± 0.03 ^Ce^	6.15 ± 0.05 ^Dd^	7.24 ± 0.06 ^Bb^	7.93 ± 0.08 ^Ab^	2.3 × 10^−9^
R150	*E. faecium*	7.28 ± 0.08 ^Bc^	7.08 ± 0.08 ^Bb^	7.22± 0.15 ^Bb^	7.79 ± 0.25 ^Ab^	0.0091

Equal capital letters in the row and equal lowercase letters in the column indicate means that are not statistically different (*p* ≥ 0.05).

## Data Availability

The authors confirm that the data supporting the findings of this study are available within the article.

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
