# Peer review of "Multidrug Resistance in Enterococci Isolated from Cheese and Capable of Producing Benzalkonium Chloride-Resistant Biofilms"

_biology, 2023, doi:10.3390/biology12101353_

Round 1

Reviewer 1 Report

Dear Authors

I hope this letter finds you well. I would like to praise you on the research presented in your manuscript titled "Multidrug resistance in enterococci isolated from cheese and capable of producing benzalkonium chloride resistant biofilms". I have carefully reviewed the manuscript and found it to be well-structured and supported by robust data. However, I believe that a few corrections and suggestions for improvement could enhance the clarity and impact of your findings. Please consider the following points:

Methodology and Data Collection: Could you provide more details on the methodology used for collecting the cheese samples and isolating Enterococcus faecium and Enterococcus faecalis strains? Specifically, how were the samples collected, and were there any specific criteria for sample selection?

Identification of Microorganisms: Did you perform phylogenetic analysis to confirm its identity as Enterococcus faecium and Enterococcus faecalis strains? Also, I am not able to find out the accession no. of submitted nucleotide sequences of 16S rRNA and other genes. Please provide the data.

Virulence Genes and Public Health Implications: Given that virulence genes were found in the isolates, could you elaborate on the potential public health implications of this finding?

Data Representation: It would be beneficial for readers if you could include backronyms (full form) for the abbreviations used in the table in a legend placed below the table. This practice will greatly facilitate understanding and interpretation of the data, ensuring that readers can easily grasp the meaning of the abbreviations without any ambiguity. Tables and Figure should be self-explanatory. Do it for all the tables and figure.

Overall, the manuscript represents a valuable contribution to the field of food microbiology and food safety. These questions aim to provide a more comprehensive understanding of the topic, which can be valuable for both the scientific community and food safety practitioners.

1.      Keep uniformity in °C/ ºC (underlined º) symbol. Look for similar types of mistakes. Like CFU/ m2 (in the Discussion section; "Among the E. faecalis isolates, one of the treatments demonstrated a decrease ≥ 3 Log CFU/m2."). Keep uniformity CFU/cm2.

2.    Missing periods at the end of some sentences. Full stop missing after “Azeitão and Nisa (21 to 97%) [40]” in the Discussion section. Look for similar types of punctuation mark mistakes in the whole manuscript.

3.    Inconsistent spacing before and after parentheses and brackets.

4.    Unclosed bracket; look “Two isolates exhibited MIC ≥10 mg/L (BAC resistance [30].” In the Discussion section.

5.    Ensure consistency in using numerals or spelled-out numbers. For instance, "24 hours" could be consistently written as "24 h."

6.    Some sentences are quite lengthy and could be revised for clarity and readability. Example: Rewrite the sentence and remove unwanted punctuation mark from Result section 3.1: “In both the most frequent resistances were to vancomycin, erythromycin, and penicillin (Table 1). vancomycin was almost complete (7/8), receptively for E. Faecalis and E. Faecium.” (Unwanted full stop, species name starts with lower case)

7.    Abbreviations and Acronyms: Consider defining acronyms and abbreviations when they are first used to improve readability, especially for non-expert readers.

8.    Consistency: Check for consistent capitalization and formatting of scientific terms and names.

READ THE WHOLE MANUSCRIPT CAREFULLY AND RECTIFY THESE TYPES OF MISTAKES.

Remember that scientific writing often follows specific conventions, so it's essential to adhere to the style and formatting guidelines provided by the target journal or publication. Additionally, it's always a good practice to have someone else, such as a peer or editor, review the paper for grammar and style before submission to ensure its clarity and correctness.

Author Response

Comments and Suggestions for Authors

Dear Authors

I hope this letter finds you well. I would like to praise you on the research presented in your manuscript titled "Multidrug resistance in enterococci isolated from cheese and capable of producing benzalkonium chloride resistant biofilms". I have carefully reviewed the manuscript and found it to be well-structured and supported by robust data. However, I believe that a few corrections and suggestions for improvement could enhance the clarity and impact of your findings. Please consider the following points:

Answer: Dear reviewer, we also hope that everything is going well with you. Thank you very much for your comments and contributions to improving our manuscript. Some of the answers to your comments were posted in the revised manuscript and highlighted in yellow.

  1. Methodology and Data Collection: Could you provide more details on the methodology used for collecting the cheese samples and isolating Enterococcus faecium and Enterococcus faecalis strains? Specifically, how were the samples collected, and were there any specific criteria for sample selection?
  2. Identification of Microorganisms: Did you perform phylogenetic analysis to confirm its identity as Enterococcus faecium and Enterococcus faecalis strains? Also, I am not able to find out the accession no. of submitted nucleotide sequences of 16S rRNA and other genes. Please provide the data.

Answer: This description in the Materials and Methods section has now been improved. Line 93-125.

Virulence Genes and Public Health Implications: Given that virulence genes were found in the isolates, could you elaborate on the potential public health implications of this finding?

Answer: Thank you for your suggestion. Done. Line: 376-386

  1. Data Representation: It would be beneficial for readers if you could include backronyms (full form) for the abbreviations used in the table in a legend placed below the table. This practice will greatly facilitate understanding and interpretation of the data, ensuring that readers can easily grasp the meaning of the abbreviations without any ambiguity. Tables and Figure should be self-explanatory. Do it for all the tables and figure.

Answer: Thank you for your observation. Done.

  1. Overall, the manuscript represents a valuable contribution to the field of food microbiology and food safety. These questions aim to provide a more comprehensive understanding of the topic, which can be valuable for both the scientific community and food safety practitioners.

Answer: Thank you for your comments.

Comments on the Quality of English Language

  1. Keep uniformity in °C/ ºC (underlined º) symbol. Look for similar types of mistakes. Like CFU/ m2 (in the Discussion section; "Among the E. faecalis isolates, one of the treatments demonstrated a decrease ≥ 3 Log CFU/m2."). Keep uniformity CFU/cm2.

Answer: Thank you for your observation. Corrected

  1. Missing periods at the end of some sentences. Full stop missing after “Azeitão and Nisa (21 to 97%) [40]” in the Discussion section. Look for similar types of punctuation mark mistakes in the whole manuscript.

Answer: Thank you for your attention. Done

  1. Inconsistent spacing before and after parentheses and brackets.

Answer: Thank you for your attention. Done

  1. Unclosed bracket; look “Two isolates exhibited MIC ≥10 mg/L (BAC resistance [30].” In the Discussion section.

Answer: Thank you for your attention. Done

  1. Ensure consistency in using numerals or spelled-out numbers. For instance, "24 hours" could be consistently written as "24 h."

Answer: Done

  1. Some sentences are quite lengthy and could be revised for clarity and readability. Example: Rewrite the sentence and remove unwanted punctuation mark from Result section 3.1: “In both the most frequent resistances were to vancomycin, erythromycin, and penicillin (Table 1).

Answer: Thank you for your attention. Done

  1. vancomycin was almost complete (7/8), receptively for E. Faecalis and E. Faecium.” (Unwanted full stop, species name starts with lower case)

Answer: Done

  1. Abbreviations and Acronyms: Consider defining acronyms and abbreviations when they are first used to improve readability, especially for non-expert readers.

Answer: Thank you for your observation. Done

  1. Consistency: Check for consistent capitalization and formatting of scientific terms and names.

Answer: Thank you for your observation.

 READ THE WHOLE MANUSCRIPT CAREFULLY AND RECTIFY THESE TYPES OF MISTAKES.

Remember that scientific writing often follows specific conventions, so it's essential to adhere to the style and formatting guidelines provided by the target journal or publication. Additionally, it's always a good practice to have someone else, such as a peer or editor, review the paper for grammar and style before submission to ensure its clarity and correctness.

Answer: Thank you for your observation.

Reviewer 2 Report

The manuscript entitled "Multidrug resistance in enterococci isolated from cheese and capable of producing benzalkonium chloride resistant biofilms" is a well-written document, the findings are relevant to the field and the conclusions are supported by the findings. The methodology used is correct. I have just some minor suggestions:

Please verify the bacterial names in the entire document, they must be written italicized.

In Table 1 verify the name of the genes (FrsB must be frsB)

In page 7 verify rephrasing "...was observed in R17, whth..."? is it correct or you mean with?

Please verify the correct name o the gene vancA or vanA?

Author Response

Comments and Suggestions for Authors

The manuscript entitled "Multidrug resistance in enterococci isolated from cheese and capable of producing benzalkonium chloride resistant biofilms" is a well-written document, the findings are relevant to the field and the conclusions are supported by the findings. The methodology used is correct. I have just some minor suggestions:

Answer: Dear reviewer, thank you for your comments and contributions to improving our manuscript. Some of the answers to your comments were posted in the revised manuscript and highlighted in yellow.

  1. Please verify the bacterial names in the entire document, they must be written italicized.

Answer: Thank you for your attention. Done

  1. In Table 1 verify the name of the genes (FrsBmust be frsB)

Answer: Thank you for your attention. Done

  1. In page 7 verify rephrasing "...was observed in R17, whth..."? is it correct or you mean with?

Answer: Thank you for your attention. It was a mistake. Done

  1. Please verify the correct name the gene vancAor vanA?

Answer: Thank you for your attention. It has been changed to vanA.

Reviewer 3 Report

REVIEW Biology-2621040

Multidrug resistance in enterococci isolated from cheese and capable of producing benzalkonium chloride resistant biofilms

By Acácio Salamandane, Gomes Cahango, Belo Muetanene, Manuel Malfeito-Ferreira, Luisa Brito.

The manuscript discusses an interesting topic for the microbiology and food biopreservation. The obtained results are interesting and can find a successful practical application.

1. The Introduction part provides a good background to the topic.

However, the negative effects of enterococci (E. faecium and E. faecalis) should be also noted: they are normal inhabitants of the intestinal tract, but can cause various infections under certain conditions, including cystitis, pyelonephritis, endocarditis, abdominal and pelvic infections (polymicrobial); their presence in food products and water is an indicator of faecal contamination, etc…

2. The Materials and methods section provides adequate and detailed description of the analyses used in the study.

3. The Experimental section is well arranged and described. The results are well presented; the figures and tables are clear and informative.

4. The conclusion is adequate.

5. The references are sufficient and up-to-date.

Author Response

“Multidrug resistance in enterococci isolated from cheese and capable of producing benzalkonium chloride resistant biofilms”

By Acácio Salamandane, Gomes Cahango, Belo Muetanene, Manuel Malfeito-Ferreira, Luisa Brito.

 The manuscript discusses an interesting topic for the microbiology and food biopreservation. The obtained results are interesting and can find a successful practical application.

Answer: Dear reviewer, thank you for your comments and contributions to improving our manuscript. Some of the answers to your comments were posted in the revised manuscript and highlighted in yellow.

  1. The Introduction part provides a good background to the topic.

However, the negative effects of enterococci (E. faecium and E. faecalis) should be also noted: they are normal inhabitants of the intestinal tract, but can cause various infections under certain conditions, including cystitis, pyelonephritis, endocarditis, abdominal and pelvic infections (polymicrobial); their presence in food products and water is an indicator of faecal contamination, etc…

Answer: Thank you for your suggestion. Done Line: 70-74

  1. The Materials and methods section provides adequate and detaileddescription of the analyses used in the study.

Answer: Thank you for your comment.

  1. The Experimental section is well arranged and described. The results are well presented; the figures and tables are clear and informative.

Answer: Thank you for your comment.

  1. The conclusion is adequate.

Answer: Thank you for your comment.

  1. The references are sufficient and up-to-date.

Answer: Thank you for your comment.
